# Longitudinal Patterns in the Isolation and Antimicrobial Resistance of Bovine Mastitis-Causing Bacteria in Ireland

**DOI:** 10.3390/antibiotics14030243

**Published:** 2025-02-27

**Authors:** Amalia Naranjo-Lucena, Philip Becker, Gillian Madigan, Rebecca Cupial, Brian Byrne, Alan Johnson

**Affiliations:** 1National Reference Laboratory for Antimicrobial Resistance, Department of Agriculture, Food and the Marine, Backweston Laboratory Campus, W23 VW2C Celbridge, Ireland; 2Food Microbiology Division, Department of Agriculture, Food and the Marine, Backweston Laboratory Campus, W23 VW2C Celbridge, Ireland; 3Regional Veterinary Laboratory, Department of Agriculture, Food and the Marine, Knockalisheen, V94 WK4 Limerick, Ireland

**Keywords:** mastitis, seasonality, antimicrobial resistance, One Health

## Abstract

**Background/Objectives**: Bovine mastitis remains a challenge for the Irish dairy industry. This study aimed to explore the seasonality and antimicrobial resistance of mastitis pathogens obtained by the regional laboratories (RVL) of the Department of Agriculture, Food and the Marine. **Methods**: Seasonality of isolation of the most common bacterial species and antimicrobial resistance of those species repeatedly obtained in the same herds in different years were explored using the RVL diagnostic data. Additionally, whole genome sequencing (WGS) was employed to establish the persistency of *Staphylococcus aureus* strains within the same herd. **Results**: A clear seasonality was observed in the isolation of *Staphylococcus aureus*, *Streptococcus uberis*, and *Escherichia coli* from milk. Seasonal differences were statistically significant within and between bacterium. Persistence of resistance within herds was highest in *S. aureus* against penicillin (35.5% of herds) and in *S. uberis* against pirlimycin (14% of herds), while *E. coli* did not show persistence of resistance to any antimicrobials. Sequencing of *S. aureus* isolates revealed that the strains causing mastitis in ten out of twenty-one herds were similar genetically in different years. In seven of these herds, *S. aureus* was persistently resistant to penicillin. Isolates from two different herds were practically identical and carried the human immune evasion cluster genes (IEC, *scn*, *sak*, *chp* and *sea*) suggesting a recent human-bovine host switch event. **Conclusions**: These findings underscore the importance of implementing targeted biosecurity measures and monitoring programs to mitigate the spread of mastitis-causing pathogens and enhance antimicrobial stewardship in the Irish dairy industry, while it also highlights the significance of including a One Health perspective in surveillance programs.

## 1. Introduction

The Irish dairy sector has experienced a steady expansion in cow numbers and milk production that has been facilitated by the implementation of key technologies [1]. The dairy industry remains a cornerstone of Irish agriculture despite a slight decrease in the value of dairy-related exports in 2023 compared to 2022, and a reduction in production primarily attributed to the impact of nitrates regulations on farm stocking rates [2,3].

There are about 17,000 dairy farms in Ireland. One of the challenges experienced by the sector is bovine mastitis, defined as an inflammation of the mammary gland. Bovine mastitis can be produced by a great variety of infectious organisms [4]. In Ireland, the most common pathogens causing mastitis are *Staphylococcus aureus*, *Streptococcus uberis*, *Escherichia coli*, *Streptococcus dysgalactiae*, coagulase-negative *Staphylococci*, and other pathogens like *Bacillus* spp. or *Trueperella pyogenes* [5,6]. While bovine mastitis reduces milk production and, therefore, reduces income, there are also increased costs for the farmer, particularly due to diagnosis, treatment, and veterinary attention. Further losses are associated with discarded milk due to withdrawal periods applied to avoid antimicrobial residues [7].

The adoption of the One Health approach aims to reduce antimicrobial resistance (AMR) through surveillance and interventions in healthcare facilities, animals, food, and the environment, emphasizing their interconnectedness [8]. Surveillance studies in the EU have estimated a decrease in antimicrobial use in food-producing animals by 44% between 2014 and 2021 [9]. European regulations on veterinary medicines (Regulation (EU) 2019/6) and medicated feed (Regulation (EU) 2019/4) introduced a ban on the preventive use of antimicrobials, and restrictions on metaphylactic use which came into effect in January 2022. However, blanket use of dry cow therapy was still common in 2019 and while the overall use of intramammary tubes for in-lactation therapy or dry cow therapy showed a decrease, the use of tubes containing critically important antimicrobials (CIA, according to WHO classification) has increased [10]. The need to reduce the use of antimicrobial agents in animal husbandry has been emphasized in an EMA and EFSA Joint Scientific Opinion from 2017 [11]. This report advocates for the “Reduce, Replace, Rethink” approach: Reduce antimicrobial usage, Replace antimicrobials with alternative treatments, and Rethink livestock production systems [11].

This study aimed to evaluate the seasonality of the frequency of isolation, phylogenetic characteristics, and antimicrobial resistance of mastitis-causing pathogens, based on data collected by the Department of Agriculture, Food and the Marine’s (DAFM) Regional Veterinary Laboratories (RVL) between the years 2020 and 2023.

## 2. Results

### 2.1. Seasonality

After removing samples with mixed growth or sterile culture results, a total of 767 (2020), 701 (2021), 936 (2022), and 848 (2023) individual isolates were identified for further analysis. From these, *E. coli*, *S. aureus*, *S. dysgalactiae*, and *S. uberis* represented the most common bacterial species isolated (Table 1). Other bacterial species obtained included other *Staphylococcus* spp., *Bacillus* spp., *Enterococcus* spp., other *Streptococcus* spp., etc.

Statistical power analysis indicated that, with a significance level of 0.05 and an effect size of 0.2, the seasonal analysis achieved a power of 0.92 for winter, 0.95 for spring, 0.94 for summer, and 0.94 for autumn, based on data aggregated across all years. The seasonality patterns found were similar every year in all bacterial species (Figure 1) with no significant statistical differences being detected between different years within the same bacterial species. *E. coli* was most prevalent in late winter and early spring, peaking between January and April in all years analyzed. *E. coli* was least prevalent in late summer and early autumn. In fact, there were significant differences between summer and winter and autumn and winter (*p* = 0.0004). On the other hand, *S. aureus* was most prevalent during late summer and autumn in all years, peaking in September in three out of four of the years analyzed, while its lowest prevalence was recorded in late winter and spring, between January and April. Significant differences were found between summer and winter (*p* = 0.03), autumn and winter (*p* = 0.008), and autumn and spring (*p* = 0.04) for *S. aureus*. *S. uberis* reached its peak prevalence in July-August every year, and significant differences were found between summer and winter (*p* = 0.002). Finally, *S. dysgalactiae* showed a low prevalence and non-significant seasonality differences overall (Kruskal–Wallis chi-squared = 2.74, *p* = 0.4343).

Generalized linear models (GLM) with Tukeys Honest Significant Difference (HSD) were employed to analyze overall seasonal differences between species (Table 2). *S. uberis* and *S. aureus* were the only pair showing no significant differences in any season. *E. coli* did not show significant differences in spring with *S. aureus* or *S. uberis*. All other comparisons were significantly different.

Peak-to-low ratio results show these same trends, confirming a degree of seasonal variation in the prevalence of each bacterial species (Table 3). The year 2022 shows a more pronounced seasonal variation in all three bacterial species, with higher ratios obtained.

### 2.2. Descriptive Longitudinal Analysis of Antimicrobial Resistance

For this analysis, only those herds that had samples submitted for two or more years, were appropriate to be tested, and from which a pure culture of *E. coli*, *S. aureus*, *S. uberis*, or *S. dysgalactiae* was obtained were selected. This resulted in a cohort of 190 herds out of the 1762 that submitted samples during that time. In 40 of these herds, more than one pathogen was detected as the cause of mastitis. *E. coli* was isolated from 55 herds, *S. aureus* from 76, *S. uberis* from 92, and *S. dysgalactiae* from seven herds. Given the descriptive nature of this analysis, no statistical power calculation or inferential analysis was conducted.

Table 4 shows the results of the number of herds from which resistant isolates to relevant antimicrobials were obtained and the persistence of this resistance in different years (i.e., if resistance to a particular antimicrobial was recorded in several years in the same bacterial species). Overall, *E. coli* isolates were most resistant to tetracycline, streptomycin, and ampicillin (30.91%, 25.45% and 23.63% of herds, respectively). Resistance to these antimicrobials was not persistent in any herd (i.e., herds from which a resistant *E. coli* was observed against any antimicrobial, only showed this resistance once). *S. aureus* resistant to penicillin was obtained in more than half of the herds (n = 47–61.84% of herds) from which *S. aureus* was isolated for several years. Resistance was maintained for two or more years in isolates from 27 herds (35.53%). *S. uberis* was most resistant to pirlimycin and erythromycin, which was present in isolates from 45 (48.91%) and 29 (31.52%) herds, respectively. In this case, persistence of resistance was observed in 13 (pirlimycin) and 2 (erythromycin) herds. *S. uberis* resistant to both pirlimycin and erythromycin were present in 23 herds. Finally, *S. dysgalactiae* that was intermediate or fully resistant to tetracycline was obtained from seven different herds, with three of them showing this resistance in two or more years.

### 2.3. Phylogenetic Analysis of S. aureus Isolates

The most common spa types observed were t524 and t529 (Table 5). A dendrogram or UPGMA tree was produced using seven allelic differences to visualize the relationship of isolates within farms (see Appendix A). Four different thresholds of single linkage cluster (SLC) were determined and the number of allelic differences according to the levels SLC1, SLC5, SLC7, and SLC24 are indicated in Table 5.

The analysis of allelic differences using core genome multi-locus sequence typing (cgMLST) resulted in the generation of 11 clusters (defined as a maximum of 24 allelic differences), with two to three isolates in each one (Figure 2). 

Ten minimum spanning tree (MST) clusters were formed by isolates from the same herd, including cluster one which was formed by the two isolates from herd number 10 (obtained in 2022 and 2023) and one from herd number 24 (obtained in 2021). The latest isolate, however, showed lower similarity than the two from herd number 10. Seven of the ten herds belonged to group two (resistant to penicillin over time), and three to group one (susceptible to penicillin over time). Isolates from herds with variable resistance did not cluster together.

Cluster number three was the exception, formed by two isolates from different herds (8 and 12) in the same year that were virtually genetically identical, showing differences in only one to two alleles. Figure 2 shows the MST formed by all 43 isolates, with the 11 clusters highlighted.

### 2.4. Phenotypic Resistance, AMR Genes, and Virulence Factors

Phenotypic resistance was determined by minimum inhibitory concentration (MIC) for all 43 *S. aureus* isolates. All isolates that were susceptible to penicillin (n = 17), except for one, shared the same spa type (t529). Isolates that were resistant to penicillin carried the *blaZ* gene and were classified into different spa types: t002 (two isolates), t8892 (four isolates), t2734 (two isolates), t224 (six isolates), t10926 (one isolate), t524 (ten isolates), and t230 (one isolate). Resistance to penicillin was significantly associated with the persistence of strains in a herd (*p* = 0.037). Resistance to other antimicrobials was also determined by MIC: one isolate was resistant to chloramphenicol and carried the *fexA* gene, one isolate was resistant to tetracycline and carried *tet(38)* gene, two isolates were resistant to trimethoprim and one isolate was resistant to sulfamethoxazole. Resistance genes were not found for trimethoprim and sulfamethoxazole. Other resistance genes like *fosB*, *lmrS* or *mepA* were present but resistance was not observed phenotypically (see Appendix A for all resistance and virulence genes data).

Regarding virulence genes, all isolates possessed the genes aur (aureolysin); cap8A, cap8B, cap8C, cap8D, cap8E, cap8F, cap8 G, cap8L, cap8LM, cap8N, cap8O, cap8P (capsular polyssacharides); esaA, esaB, esaD, esaE, esaG1, esaG2, esaG4, essA, essB, ess C, esxA, esxB, esxC, esxD (secretion systems), hld, hlgA, hlgB, hlgC, hly/hla (hemolysins), isdA, isdC, isdD, isdE, isdF, isdG (iron surface determinants), lukD (leukocidin), srtB (sortase); sspB and sspC (Staphopain and spathostatin). Other genes were less common across all bacterial isolates (Appendix A). No significant association was found between virulence genes and the persistence of strains in a herd. The presence of genes sak (staphylokinase), sea (enterotoxin), and host immune evasion genes chp and scn (staphylococcal complement inhibitor) was specific to spa type t2 isolates (isolates from herds 8 and 12 forming MST cluster three).

## 3. Discussion

Mastitis is a challenge to Irish dairy farmers with costs of about 60 EUR/cow/year derived from milk production losses, diagnostic testing, treatment, discarded milk, veterinary attention, culling, etc. [12,13]. Traditionally, blanket dry cow therapy (DCT), which treats all animals with antimicrobials at drying off regardless of their infection status, has been used widely in Ireland [14]. New approaches like selective DCT are encouraged due to concerns about the use of antimicrobials on farms and the introduction of the new Veterinary Medicines legislation (Regulation (EU) 2019/6) to prevent the development of AMR. However, this poses new challenges, particularly in herds with suboptimal mastitis control, and a better understanding of the epidemiology of mastitis pathogens present in a particular farm is, therefore, key [15]. We found that the seasonality patterns and antimicrobial resistance results obtained in previous years on the same herd can be useful tools for private veterinary practitioners (PVP) to decide on appropriate therapeutic treatment of mastitis cases. This is particularly important in acute or severe cases where action is needed under animal welfare grounds before any laboratory results can be obtained.

Most Irish dairy herds practice spring calving, with most of the dairy calves being born in the first six months of the year [16]. This is coordinated so that the peak of milk production occurs during optimal grass growth, allowing for a primarily pasture-based production where 81–83% of the diet consists of grass [17]. A marked seasonality in the isolation of the three main pathogens causing bovine mastitis, *E. coli*, *S. uberis*, and *S. aureus*, was observed. This was particularly true for the year 2022, and the reasons for this may be varied and include climatological, herd management, and bacterial characteristics. As an opportunistic invader, *E. coli* can survive in a contaminated environment, particularly under warm and humid conditions and cause mastitis if the conditions are agreeable [4]. Mastitis caused by *E. coli* is, therefore, more usually seen during the winter housing period, when there is more opportunity for contact with housing elements, fecal material, etc. In addition, cows are particularly susceptible during the peri-partum period due to increased metabolic and physical stress [18,19]. It is, therefore, not surprising to observe an increased prevalence of *E. coli* mastitis during the winter housing period. Furthermore, *E. coli* has been shown to survive in the mammary gland for long periods, resulting in repeat cases of mastitis [20]. This can also explain cases during early spring in all years, as shown by our results.

On the other hand, *S. uberis* is also considered an environmental pathogen, being associated in particular with straw-bedded yards [18]. However, unlike *E. coli*, we observed that the isolation of *S. uberis* was higher during the summer months. This is in agreement with a Dutch study which found *S. uberis* incidence in mastitis cases to be highest during August [21]. *S. uberis* was lower in totally confined herds than in herds that pastured the cows. In fact, in New Zealand, where cows are on pasture all year round, the most common mastitis pathogen is *S. uberis* [22]. Environmental exposure to soil, pasture and water during the summer months may be more relevant in *S. uberis* mastitis. Additionally, there is evidence that direct transmission between cows can also happen particularly during the milking process [23,24]. The increased number of mastitis cases caused by *S. uberis* may, therefore, originate from a combination of both environmental and contagious factors. Infections by *S. dysgalactiae* did not show seasonal variation, although low numbers were available overall.

Mastitis infection caused by *S. aureus* is probably considered the greatest challenge because cure rates using antibiotics are low while infection is usually subclinical and can become chronic, making it difficult to control [19,25,26]. According to our results, *S. aureus* mastitis was more prevalent during late summer and autumn. *S. aureus* is considered a contagious pathogen, with transmission between animals happening via the milking machine, udder cloths, milker’s hands, etc. [27,28]. It is for this reason that infection with *S. aureus* happens most commonly during the milking period. Furthermore, Staphylococci colonize damaged skin and teat lesions which are more commonly found in late lactation [29,30].

We acknowledge that this study does not include data on herd characteristics, management practices, bedding types, or other factors that could influence the occurrence of bovine mastitis by different bacterium. As such, these factors were not accounted for in the analysis, which may limit the ability to fully interpret the seasonality observed. However, we believe these results are relevant from a biosecurity point of view; herds with consistent mastitis problems during the summer months are advised to control fecal contamination of their pastures, thereby reducing the presence of *S. uberis*. This can be achieved by rotating grazing areas, avoiding overstocking, and ensuring cows have access to dry resting areas. Routine maintenance of the milking machine, including checking vacuum levels and replacing worn components, and implementing effective post-milking disinfection protocols, such as the use of teat dips with proven efficacy, will also help reduce the incidence of mastitis caused by *S. aureus* [15,31]. On the other hand, herds experiencing mastitis issues during the winter are more likely affected by *E. coli* and are advised to focus on maintaining clean and dry bedding, ensuring proper ventilation in housing facilities, and minimizing overcrowding [18]. Peri-parturient stress, which often occurs around calving due to metabolic and physiological changes, can compromise the immune system, increasing susceptibility to infections. To mitigate this, herds should prioritize stress reduction strategies, such as providing sufficient space, minimizing disruptions, and ensuring high-quality nutrition during the transition period [32].

In this study, we also aimed to explore the resistance patterns in those farms from which the same bacterial species were obtained over several years. Although the number of farms from which *S. dysgalactiae* was isolated in several years was low (n = 7), it is relevant to highlight that all farms had isolates resistant to tetracycline at some stage, and three showed persistent resistance. This is in line with what has been observed in Ireland [6], where *S. dysgalactiae* is most commonly resistant to this antimicrobial. However, the numbers tested were low when compared to other bacterial species. The most common resistance patterns observed in the remaining bacteria were *E. coli* against tetracycline (31% of farms had a resistant isolate at some stage), *S. aureus* against penicillin (62% of farms), and *S. uberis* against pirlimycin (49% of farms). Again, these results are not unusual, although *E. coli* would be expected to have slightly greater resistance against ampicillin as observed in other European countries [33,34].

The heterogeneity of environmental *E. coli* is high, so reinfections would be expected to be caused by different strains, which is likely why our results indicate a lack of persistence of resistance in these farms. Our results, however, show that the persistence of resistance over the years in the same farm is high in *S. aureus* against penicillin (35.5% of farms), followed by pirlimycin in *S. uberis* (14% of farms).

It is crucial to perform culture and antimicrobial sensitivity testing of milk samples to guide targeted therapy and reduce the unnecessary use of antibiotics. However, in acute cases of mastitis, where immediate intervention is required, treatment often needs to be initiated before laboratory results become available. In such situations, the choice of antimicrobial should be guided by historical farm-level sensitivity patterns and knowledge of likely pathogens, while laboratory confirmation remains essential to refine ongoing treatment and inform future decisions [35]. Taking into account the seasonality and AMR findings from this study can be useful tools for PVP dealing with severe mastitis cases. For instance, a mastitis infection during the summer months is more likely caused by *S. uberis* or *S. aureus*. Therapy can, therefore, be better informed with a more targeted approach. In addition, if this infection is not responsive to therapeutic treatment with penicillin or other early-generation cephalosporin, the etiological agent will more likely be *S. aureus*.

To explore the *S. aureus* strains causing mastitis over the years, WGS of 43 isolates from 21 herds was performed to explore the phylogeny of the strains within each herd. This analysis revealed that highly similar strains repeatedly caused mastitis in 10 out of 21 herds (according to 24 allelic differences). It has been previously shown that the persistence of *S. aureus* mastitis is linked to the presence of the *blaZ* gene, which confers resistance to penicillin [36]. In 7 out of 10 herds, *S. aureus* was persistently resistant to penicillin, with the association between resistance to penicillin and strain similarity being statistically significant. We found the presence of several genes related to immune evasion (*cap8*, *isd*, *lukD*), tissue damage and inflammation (*hld*, *hla*, *hlg*), iron acquisition (*isd* system), or bacterial persistence (*aur*, *sspB*, *sspC*) that are key for the pathogenesis of *S. aureus* infections [37]. However, virulence genes that were associated with persistent infections according to Haveri et al., (2007) (*sed* and *sej*) were not present in the strains from this study. In addition, we did not find any virulence genes to be significantly associated with the persistence of strains in a herd [36].

While exploring transmission between herds was not the aim of this study, cluster number 3 was formed by two strains from different herds that were practically identical genetically. Both isolates were obtained in 2022, and the two herds happened to be 8 km apart, suggesting that transmission may have occurred between farms by farm staff/veterinarians/visitors. These two strains belonged to spa type t002, sequence type (ST) 5, carried the resistance gene *blaZ*, and carried the immune evasion cluster (IEC) virulence factors *chp*, *sak*, sea and *scn*, not present in any other strain of this study. These characteristics indicate these strains may have a human origin; ST5 is more common in humans while IEC carries human-specific innate immune modulators that are not commonly advantageous in animals [38]. Host-switch events are not rare, and have previously been documented [39,40,41]. ST5 *S. aureus* methicillin-susceptible *S. aureus* (MSSA) human isolates can exhibit a high prevalence of the IEC and demonstrate great virulence both in vitro and in vivo studies [42]. Hypervirulent human strains can jump to bovines and then lose the virulence factors relevant to the invasion of human cells [43]. The conservation of the IEC in these strains, therefore, possibly suggests a recent host-switch event.

This study utilizes data from routine diagnostic submissions which are used as a source of passive surveillance for antimicrobial resistance (AMR) in mastitis pathogens in Ireland. Antimicrobial sensitivity results are published annually in the All Island Disease Surveillance Report, which can be found at https://www.animalhealthsurveillance.agriculture.gov.ie, accessed on 23 February 2025. While there is currently no legislation mandating the monitoring of AMR in clinical veterinary isolates, several initiatives, such as JAMRAI 2 and ENOVAT, are working towards standardizing and harmonizing testing and reporting protocols across the EU. These efforts aim to develop a comprehensive EU-wide AMR surveillance program. Importantly, the data generated from this testing will contribute to JAMRAI 2, providing valuable insights into resistance trends and supporting the development of such frameworks.

## 4. Materials and Methods

### 4.1. Sample Source and Microbiological Laboratory Methods

This study relies on diagnostic samples submitted voluntarily to veterinary laboratories, which may be a limitation regarding the representativeness of the findings. Diagnostic samples are typically submitted when clinical symptoms are evident, and as such, they may over-represent cases with more severe or obvious manifestations of mastitis, potentially biasing the pathogen prevalence data. Additionally, farms with higher mastitis awareness or better access to veterinary services may be more likely to submit samples, further affecting the generalizability of the results. Therefore, the findings should be interpreted with caution.

A total of 7833 milk samples from individual bovine animals from 1762 dairy farms (about 10% of total dairy farms in Ireland) with suspected clinical or subclinical mastitis were submitted to the DAFM RVL network for diagnostic purposes between 2020 and 2023. Samples were tested if deemed appropriate by laboratory analysts. The criteria followed for accepting a sample were: (i) the sample has been obtained from an individual animal (i.e., is not a bulk tank milk (BTM) sample); (ii) the sample does not contain visible dirt or fecal material on receipt to the lab; (iii) the sample does not contain preservative tablets for somatic cell count (SCC); (iv) the sample has been collected in an appropriate container (i.e., a sterile tube or pot). All samples were tested using the California mastitis test (CMT) as a preliminary tool to confirm inflammation of the udder. Microbiological culture and antimicrobial susceptibility testing was performed on samples with scores: trace, +1, +2 and +3, which can detect SCC from 150,000–500,000 (trace) to >5 million (+3) [44].

To obtain bacterial isolates, a sterile loop with 10 µL of milk was inoculated onto three different culture media plates: Blood agar, McConkey agar and Edward’s agar (Fannin L.I.P, Galway, Ireland) to isolate, identify, and differentiate various bacteria commonly associated with bovine mastitis. The streak technique was used to obtain pure colonies and plates were incubated for 48 h at 37 °C. Growth results were recorded at 24 and 48 h. Samples that showed moderate to heavy growth of at least three different colony types were considered contaminated and testing was stopped. Suspect colonies were sub-cultured and examined with the aid of a range of additional tests such as a gram stain, oxidase, catalase and streptococcal grouping kit. The appropriate analytical profile index (API^®^ Biomérieux, Marcy-l’Étoile, France) test was then employed for genus and species identification. Isolates were then stored in cryobeads (Technical Service Consultants Ltd., Heywood, UK), at −20 °C.

### 4.2. Antimicrobial Susceptibility Testing

For the purpose of this study, the four most commonly isolated bacteria were included: *Staphylococcus aureus*, *Streptococcus dysgalactiae*, *Streptococcus uberis*, and *Escherichia coli*. Antimicrobial susceptibility testing was carried out on all pure isolates obtained using the disk diffusion method following standard guidelines [45]. Once a pure culture was obtained, each bacterial species was tested using a selected panel of antimicrobials relevant for therapy and veterinary prescription. Paper disks were obtained from Oxoid^TM^ (Thermo Scientific™ Sensititre, Waltham, MA, USA). See Table 6 for information on antimicrobials tested.

First, the inoculum was prepared by selecting two to four colonies from a pure culture of the bacteria in a non-selective agar plate which was then emulsified in 5 mL of Sensititre demineralized water (Thermo Scientific™ Sensititre, Waltham, MA, USA). The suspension was vortexed and adjusted to a density of approximately 0.5 McFarland standard (Serosep, Limerick, Ireland). A sterile swab was dipped in the suspension and then a rotary plate spreader was used to inoculate Mueller–Hinton agar plates (Fannin L.I.P, Galway, Ireland) for *S. aureus* and *E. coli* and Mueller–Hinton agar plates supplemented with 5% defibrinated sheep blood (Fannin L.I.P, Galway, Ireland) for *S. uberis* and. *S. dysgalactiae*. Quality control strains *E. coli* ATCC25922, *S. aureus* ATCC25923, or *Streptococcus pneumoniae* ATCC49619 were included in every run to confirm growth and inhibition zone sizes were within range for every antimicrobial.

To avoid the overlapping of inhibition zones and interaction between agents, a maximum of six disks were tested on a 90 mm plate for *E. coli* and *S. aureus*, while a maximum of four was used for *Streptococcus* spp. [45]. Disks were applied to the surface of the agar and plates were placed in the incubator at 35 °C (in aerobic conditions for *S. aureus* and *E. coli* and 7% CO_2_ for *Streptococcus* spp.) within 15 min. Plates were incubated for 16–18 h for *E. coli* or 20–24 h for the remaining bacteria. Calibrated callipers were used to measure inhibition zones to the nearest millimeter. Quality control strain plates were first checked for inhibition zones to be within accepted ranges, and then the test bacteria inhibition zones were measured. Clinical breakpoints available in CLSI VET01S, 7th Edition were employed to determine if isolates were susceptible, intermediately resistant or fully resistant to the antimicrobials tested.

For confirmation and surveillance purposes, isolates showing resistance to any antibiotics were further identified by Maldi-ToF (Bruker, Billerica, MA, USA) and additionally tested using broth microdilution. The method employed for testing of *E. coli* was previously described by Byrne et al., 2024 [46] Testing of *S. aureus* and *Streptococcus* spp. was performed using the same method but a different plate; Sensititre™ EU Surveillance *Staphylococcus* EUST (or EUST2) Antimicrobial Sensitivity Test (Thermo Scientific™, Waltham, MA, USA). For *S. aureus*, 10 μL of 0.5 McFarland standard (Serosep, Limerick, Ireland) in Mueller–Hinton broth (Thermo Scientific™ Sensititre, Waltham, MA, USA) was added to each well and the plate was incubated at 35 °C for 22–24 h. For *Streptococcus* spp., 50 μL of 0.5 McFarland standard in Mueller–Hinton broth supplemented with lysed horse blood (Thermo Scientific™ Sensititre, Waltham, MA, USA) was added to each well and the plate was incubated at 35 °C for 24 ± 2 h. Epidemiological breakpoints (ECOFF) as indicated by the European Committee on Antimicrobial Susceptibility Testing (EUCAST) were employed for the interpretation of results. If these were not available, EUCAST or CLSI clinical breakpoints were employed [45].

### 4.3. Data Analysis and Statistics

#### 4.3.1. Seasonality

One isolate per herd and submission date were employed for this analysis. This was done to remove the effect on the final proportions of herds submitting several samples on the same date resulting in the same bacterial species being isolated. If two different bacterial species were found to be the cause of mastitis in a particular herd at the same time, both were included. The proportion of the most common bacteria *E. coli*, *S. aureus*, *S. dysgalactiae*, and *S. uberis*, each month was calculated. Sterile and mixed cultures were excluded from the analysis.

Given the nature of the data, lower levels of submissions were received in certain months. Using a 90% confidence level and 10% margin of error, with a prevalence of 20% (for the three most common bacterial species), the minimum sample size should be 43 samples per month using the formula for sample size in proportion studies [47]. The final average monthly sample numbers were 64 in 2020, 58 in 2021, 78 in 2022, and 71 in 2023. Months where the number of samples was below 43 were December and January in 2020, 2021, 2022 and 2023, and September in 2020. For this purpose, seasons were classified as; winter: January to March, spring: April to June, summer: July to September, and autumn: October to December. Although traditional Irish seasons vary from this classification, December and January are months in which lower samples are generally received by the diagnostic laboratories, as we have observed. This can introduce variability and reduce the statistical power of the analysis, potentially biasing seasonal trends; to mitigate this, these months were distributed across different seasons (e.g., winter: January–March, autumn: October–December) to ensure their effects are balanced within the seasonal averages. This approach ensures that each season is represented by an adequate sample size, making the analysis more robust and reliable. In addition, power calculation by season using a significance level of 0.05 and an effect size of 0.2 and the two-proportion Z-test method was performed. This effect size reflects a meaningful and biologically relevant difference in bacterial prevalence.

The intensity of seasonal occurrence was estimated using a direct calculation of the peak-to-low ratio and corresponding 95% confidence intervals. Seasonal averages were calculated by grouping monthly incidence percentages into seasons. For each species and year, the average incidence percentage within each season was calculated. A population size per season of 204 was used because the average total number of isolates per month was 68. A fixed population size per season was used to reflect the proportion of each species in relation to the overall dataset and to ensure that comparisons across seasons and years are standardized. Peak-to-low ratios were calculated simply by dividing the highest seasonal average by the lowest. Peak-to-low ratios were not performed for *S. dysgalactiae* data, as average seasonal isolate numbers were very low (n = 16) and ratios may not reflect meaningful seasonal trends.

Seasonal average data for all bacterial species and by individual species were tested for normality using the Shapiro–Wilk normality test, and for homogeneity of variance using Bartlett test of homogeneity of variances. ANOVA was used to test for differences between seasons and bacterial species if assumptions were met. Tukey multiple comparisons of means was used as a post-hoc test. Otherwise, the Kruskal–Wallis rank sum test or GLM with a quasibinomial family, followed by Tukey’s HSD test for post-hoc comparisons for more complex comparisons, were employed to test for differences if the data did not meet normality and homogeneity assumptions. *p* values of <0.05 were considered to be statistically significant.

Data management and statistical analyses were performed using R version 4.3 (packages: FSA_0.9.6, tidyr_1.3.1, ggplot2_3.5.1, emmeans_1.10.6 dplyr_1.1.4 and readxl_1.4.3) and Microsoft Excel version 16.85.

#### 4.3.2. Antimicrobial Resistance Persistence on Individual Farms

For the purpose of this study, only samples from those herds from which samples were submitted over two or more years were considered appropriate for testing and a pure culture was obtained were selected to explore the persistence of resistance. If the same bacterial species was obtained in two different years, the herd was included in this analysis. This resulted in a total of 190 herds. A descriptive analysis of the resistance patterns found in these herds was performed.

### 4.4. Whole Genome Sequencing and Bioinformatic Analysis

As *S. aureus* presented the greatest level of persistence of resistance, whole genome sequencing and phylogenetic analysis were performed on 43 isolates from 21 herds to establish if the same strains were persistent within the same herd over a two or more-year period. *S. aureus* represented the different resistance patterns found against penicillin between 2020 and 2023: group 1, 12 isolates from six herds showing susceptibility to penicillin over time (herds number 1, 2, 11, 19, 22, and 23); group 2, 21 isolates from ten herds showing resistance to penicillin over time (herds number 5, 6, 7, 10, 13, 14, 15, 17, 20, and 24); and group 3, ten isolates from five herds showing a change from susceptible to resistant or vice versa to penicillin over time (herds number 3, 8, 9, 12, and 16). Appendix A contains information from these isolates.

#### 4.4.1. DNA Extraction

*S. aureus* isolates were recovered from frozen stocks on Columbia Agar with 5% Horse blood (E and O Laboratories LTD, Bonnybridge, UK) and incubated for 24 h at 37 °C under aerobic conditions. Using a 1 μL inoculation loop, a loopful of pure culture was taken from the plate and re-suspended in 100 μL lysozyme (Sigma-Aldrich, St. Louis, MO, and Burlington, MA, USA), vortexed and incubated at 56 °C for 30 min, with vortexing at 15 min. A volume of 100 μL MagNA Pure 96 bacterial lysis buffer (Roche Diagnostics, Dublin, Ireland) and 20 μL proteinase k (Roche Diagnostics, Dublin, Ireland) was added and vortexed for 20 s. Samples were incubated at 65 °C for 10 min followed by 95 °C for 10 min. DNA was extracted using the MagNA Pure 96 system (Roche Diagnostics) using MagNA Pure 96 DNA and Viral NA Small Volume Kit (Roche Diagnostics, Dublin, Ireland). The manufacturer’s protocol was followed, and the final elution volume was 100 μL.

#### 4.4.2. Library Preparation

Sample libraries for all isolates were prepared using the Illumina DNA Prep kit (Illumina, Inc., San Diego, CA, USA) as per manufacturer instructions using 30 μL of extracted DNA, and the Hamilton NGS STAR robotic liquid handler. The tagmented DNA was amplified in a working volume of 50 μL with the following settings: heated lid at 100 °C, initial cycle at 68 °C for 3 min followed by 98 °C for 3 min and 5 cycles of (98 °C for 45 s, 62 °C for 30 s and 68 °C for 2 min) with a final run at 68 °C for 1 min followed by a hold temperature of 10 °C (Hamilton, Rena, NV, USA, NGS STAR on-deck thermal cycler).

#### 4.4.3. Normalization, Denaturing, and Sequencing of Libraries

To achieve optimal cluster density, equal library volumes (5 μL) of individual libraries were pooled into a sterile 1.5 mL tube and the pool was quantified in triplicate by Qubit fluorometric quantitation using the Qubit dsDNA HS Assay Kit (Thermo Scientific™ Sensititre, Waltham, MA, USA) before sequencing. This protocol generated a library fragment size of 600 bp. The pooled libraries were diluted in resuspension buffer with Tween20 (RSB; Illumina San Diego, CA, USA) to a concentration of 2 nM. Libraries were then further diluted in RSB with Tween20 (Illumina Inc., San Diego, CA, USA) from 2 nM to give a final loading concentration of 750 pM. The sequencing was performed on a NextSeq2000 (Illumina) using a P1 300-cycle (2 × 150 bp) cartridge and P1 flow cell (Illumina, Inc., San Diego, CA, USA) with paired-end reads. The pool was spiked with 2% PhiX (Illumina, Inc., San Diego, CA, USA).

#### 4.4.4. Analysis of Whole Genome Sequencing Data

To check that the sequencing run quality passed basic quality metrics for raw sequence data, it was confirmed that >85% of the sequencing reads were greater or equal to Q30. In addition, the error rate (≥6%) and sequencing yield were assessed. The quality of the Fastq files was assessed using FASTQC v0.12.1. The fastq files for each sample were assembled using SPAdes version (4.0.0) and the Ridom SeqSphere+ (v10.0.2, Ridom GmbH, Münster, Germany) bioinformatic pipeline. Mash was used to detect contamination in sample sequences. Seqsphere+ assigned a multilocus sequence type (MLST), core genome MLST (cgMLST) [48], an AMR profile [49,50], spa type [51] and virulence profiles [50] to each sample. The allelic differences (AD) observed between the cgMLST profiles of the samples were determined by SeqSphere. The software generated single linkage clusters using thresholds of 1, 5, 7 and 24 allelic differences (AD).

An MST was built using a maximal cluster distance of 24 allele differences. Virulence factors and AMR genes were obtained via Seqsphere+ using the Virulence Factor database (VFDB, http://www.mgc.ac.cn/VFs/, accessed on 23 February 2025) and AMRfinder tool. A heatmap to visualize the virulence genes present was built using R (v4.4.0).

A chi-square test for independence was conducted in R (v4.4.0) to determine whether herd penicillin susceptibility patterns (susceptible, resistant, or variable over time) were associated with the persistence of the same strain causing mastitis. Similarly, the association of virulence genes with persistence was explored. Statistical significance was defined as *p* < 0.05.

## 5. Conclusions

We found that there is a clear seasonality in the agents causing mastitis in Ireland. In addition, we established that resistance to penicillin in *S. aureus* is associated with the persistence of a strain causing mastitis in a herd. These results can help understand the epidemiology of this disease and are key to implementing successful control measures. While this study did not aim to explore between herds transmission, cgMLST analysis indicated two strains from different herds were highly similar, and carried human-specific genetic determinants which may indicate a recent human transmission. This highlights the importance of including One Health perspectives in the surveillance of infections and antimicrobial resistance. Additionally, ongoing surveillance of resistance patterns and virulence factors in clinical isolates is crucial to inform effective control strategies and antimicrobial use policies.

## Figures and Tables

**Figure 1 antibiotics-14-00243-f001:**
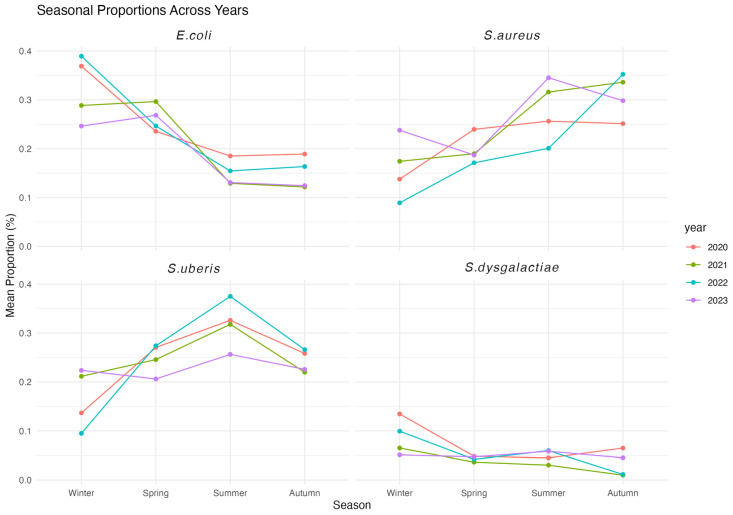
Seasonal proportions of isolation of bacterial isolates from mastitis milk samples between 2020 and 2023.

**Figure 2 antibiotics-14-00243-f002:**
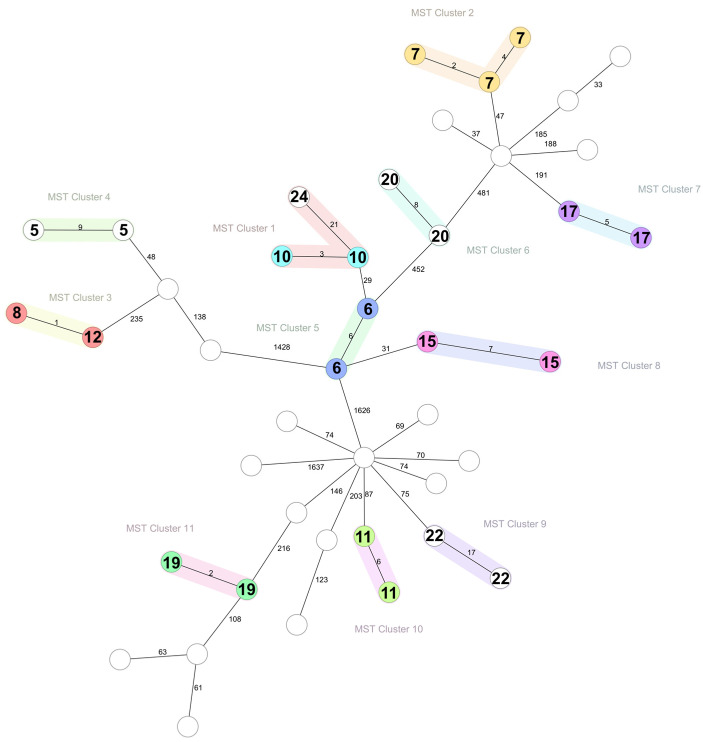
MLST minimum spanning tree of 43 *S. aureus* isolates. Clusters are shown with a maximal cluster distance of 24 allele differences and colored. Numbers in bold indicate herd ID. Numbers at the connecting lines indicate the number of allelic differences between isolates.

**Table 1 antibiotics-14-00243-t001:** Annual proportion of *E. coli*, *S. aureus*, *S. uberis*, and *S. dysgalactiae* obtained from milk cultures by the RVL (results show percentage within total isolates obtained per year, including minor species, and number of isolates in parenthesis).

Bacteria	2020 (n = 767)	2021 (n = 701)	2022 (n = 936)	2023 (n = 848)
*E. coli*	24.77% (190)	21.83% (153)	20.67% (205)	19.93% (169)
*S. aureus*	22.69% (174)	24.25% (170)	21.37% (212)	25.59% (217)
*S. uberis*	25.68% (197)	24.39% (171)	25.2% (250)	22.64% (192)
*S. dysgalactiae*	6.91% (53)	3.57% (25)	4.44% (44)	5.31% (45)

**Table 2 antibiotics-14-00243-t002:** Comparisons of bacterial species proportions within each season.

Season	Comparison	Estimate (Log-Odds)	Std. Error	z-Value	*p*-Value
Winter	*S. aureus*–*E. coli*	−0.92	0.192	−4.79	<0.001
	*S. dysgalactiae*–*E. coli*	−1.61	0.229	−7	<0.001
	*S. uberis*–*E. coli*	−0.87	0.19	−4.58	<0.001
	*S. dysgalactiae*–*S. aureus*	−0.68	0.248	−2.76	0.029
	*S. uberis*–*S. aureus*	0.05	0.212	0.24	0.995
	*S. uberis*–*S. dysgalactiae*	0.73	0.246	2.98	0.015
Spring	*S. aureus*–*E. coli*	−0.37	0.159	−2.32	0.09
	*S. dysgalactiae*–*E. coli*	−2.06	0.254	−8.09	<0.001
	*S. uberis*–*E. coli*	−0.07	0.152	−0.44	0.971
	*S. dysgalactiae*–*S. aureus*	−1.69	0.259	−6.52	<0.001
	*S. uberis*–*S. aureus*	0.3	0.16	1.88	0.227
	*S. uberis*–*S. dysgalactiae*	1.99	0.255	7.81	<0.001
Summer	*S. aureus*–*E. coli*	0.79	0.182	4.34	<0.001
	*S. dysgalactiae*–*E. coli*	−1.24	0.276	−4.51	<0.001
	*S. uberis*–*E. coli*	0.98	0.179	5.45	<0.001
	*S. dysgalactiae*–*S. aureus*	−2.03	0.262	−7.75	<0.001
	*S. uberis*–*S. aureus*	0.19	0.157	1.19	0.623
	*S. uberis*–*S. dysgalactiae*	2.22	0.26	8.52	<0.001
Autumn	*S. aureus*–*E. coli*	0.93	0.148	6.31	<0.001
	*S. dysgalactiae*–*E. coli*	−1.65	0.263	−6.27	<0.001
	*S. uberis*–*E. coli*	0.6	0.153	3.92	<0.001
	*S. dysgalactiae*–*S. aureus*	−2.58	0.252	−10.26	<0.001
	*S. uberis*–*S. aureus*	−0.34	0.133	−2.53	0.052
	*S. uberis*–*S. dysgalactiae*	2.24	0.254	8.82	<0.001

**Table 3 antibiotics-14-00243-t003:** Seasonal peak-to-low ratios for most common pathogens (2020–2023).

	Year	Peak to Low Ratio (95% CI)	Peak	Low
*E. coli*	2020	1.993 (1.990–1.997)	Winter	Summer
*E. coli*	2021	2.435 (2.43–2.439)	Spring	Autumn
*E. coli*	2022	2.520 (2.517–2.524)	Winter	Summer
*E. coli*	2023	2.159 (2.155–2.163)	Spring	Autumn
*S. aureus*	2020	1.861 (1.857–1.865)	Summer	Winter
*S. aureus*	2021	1.928 (1.925–1.932)	Autumn	Winter
*S. aureus*	2022	3.952 (3.947–3.956)	Autumn	Winter
*S. aureus*	2023	1.845 (1.841–1.848)	Summer	Spring
*S. uberis*	2020	2.384 (2.38–2.388)	Summer	Winter
*S. uberis*	2021	1.502 (1.499–1.505)	Summer	Winter
*S. uberis*	2022	3.952 (3.948–3.957)	Summer	Winter
*S. uberis*	2023	1.243 (1.24–1.247)	Summer	Spring

**Table 4 antibiotics-14-00243-t004:** Number of herds from which a resistant isolate was obtained and number of herds in which resistance to that antimicrobial persisted in two or more years (shown in parenthesis).

Bacteria	Amp	Pen	AmC	Fox	Fur	Pod	Enr	Kan	Str	Ery	Pir	Sxt	Tet
*E. coli* (n = 55)	13(0)	-	2-1i(0)	-	0	0	2(0)	4-1i(0)	14-2i(0)	-	-	4(0)	17(0)
*S. aureus* (n = 76)	-	47(27)	-	0	0	-	-	-	-	1i(0)	1(0)	1(0)	3(1)
*S. uberis* (n = 92) *	5(0)	7(0)	0	-	0	-	-	-	-	29(2)	45(13)	-	20-3i(4)
*S. dysgalactiae* (n = 7) *	0	0	0	-	0	-	-	-	-	0	0	-	5-2i(3)

Amp, ampicillin; Pen, penicillin; AmC, amoxicillin-clavulanic acid; Fox, cefoxitin; Fur, ceftiofur, Pod, cefpodoxime; Enr, enrofloxacin; Kan, kanamycin; Str, streptomycin; Ery, erythromycin; Pir, pirlimycin; Sxt, Sulfamethoxazole-trimethroprim; Tet, tetracycline. (i for intermediate resistance). * *Streptococcus* spp. were tested for either both B-lactam antimicrobials ampicillin and penicillin or only one of them.

**Table 5 antibiotics-14-00243-t005:** Core genome MLST, spa typing results of 43 *S. aureus* isolates from 21 herds and MST clusters with maximum allelic differences found within each SLC threshold.

Herd ID	Sample ID	Year	Spa Type	ST	MST Cluster	SLC 1 AD	SLC 5 AD	SLC 7 AD	SLC 24 AD
1	GB21-005343	2021	t529	1074	-	-	-	-	-
1	GB24-001550	2023	t529	151	-	-	-	-	-
2	GB21-005347	2021	t4494	151	-	-	-	-	-
2	GB22-007486	2022	t529	151	-	-	-	-	-
3	GB22-000385	2023	t8892	5	-	-	-	-	-
3	GB24-001093	2021	t529	151	-	-	-	-	-
5	GB22-000485	2022	t8892	5	4	-	-	-	9
5	GB22-006085	2021	t8892	5	4	-	-	-	9
6	GB22-000636	2021	t224	124	5	-	-	6	6
6	GB24-001092	2023	t224	124	5	-	-	6	6
7	GB22-002308	2021	t524	-	2	-	2	2	2
7	GB22-007365	2022	t524	-	2	-	4	4	4
7	GB24-000477	2023	t524	-	2	-	4	4	4
8	GB22-003893	2023	t002	5	3	1	1	1	1
12	GB22-004809	2023	t002	5	3	1	1	1	1
8	GB23-005571	2022	t529	-	-	-	-	-	-
9	GB22-004166	2022	t529	151	-	-	-	-	-
9	GB23-007426	2023	t524	71	-	-	-	-	-
10	GB22-004732	2022	t224	124	1	-	3	3	21
10	GB24-001053	2023	t224	124	1	-	3	3	3
24	GB24-000234	2021	t224	124	1	-	-	-	21
11	GB22-004803	2023	t529	-	10	-	-	6	6
11	GB23-005569	2022	t529	-	10	-	-	6	6
12	GB24-001196	2022	t529	-	-	-	-	-	-
13	GB22-005161	2022	t524	71	-	-	-	-	-
13	GB24-000467	2023	t524	71	-	-	-	-	-
14	GB22-005165	2022	t524	71	-	-	-	-	-
14	GB24-000550	2023	t524	71	-	-	-	-	-
15	GB22-006084	2022	t224	124	8	-	-	7	7
15	GB24-000357	2023	t10926	124	8	-	-	7	7
16	GB22-006533	2023	t230	508	-	-	-	-	-
16	GB24-001475	2022	t529	151	-	-	-	-	-
17	GB22-007358	2022	t524	71	7	-	5	5	5
17	GB24-001372	2023	t524	71	7	-	5	5	5
19	GB22-007367	2022	t529	151	11	-	2	2	2
19	GB24-000465	2023	t529	151	11	-	2	2	2
20	GB23-000304	2022	t2734	97	6	-	-	-	8
20	GB23-007429	2023	t2734	97	6	-	-	-	8
22	GB23-000457	2023	t529	151	9	-	-	-	17
22	GB23-004389	2022	t529	151	9	-	-	-	17
23	GB23-001830	2023	t529	151	-	-	-	-	-
23	GB24-001694	2022	t529	151	-	-	-	-	-
24	GB22-000482	2023	t8892	5	-	-	-	-	-

Notes: ST: Sequence type; SLC: Single Linkage Cluster; AD: Allelic Differences.

**Table 6 antibiotics-14-00243-t006:** Bacterial species and antimicrobials tested using disk diffusion. Concentration of antimicrobial in paper disks in parenthesis.

Bacterial Species	Antimicrobials
*S. aureus*	Penicillin (10 units), Cefoxitin (30 μg), Ceftiofur (30 μg), Erythromycin (15 μg), Pirlimycin (2 μg), Sulfamethoxazole-trimethoprim (23.75/1.25 μg), Tetracycline (30 μg)
*S. uberis*	Ampicillin (10 μg), Penicillin (10 units), Ceftiofur (30 μg), Erythromycin (15 μg), Pirlimycin (2 μg), Tetracycline (30 μg)
*S. dysgalactiae*	Ampicillin (10 μg), Penicillin (10 units), Ceftiofur (30 μg), Erythromycin (15 μg), Pirlimycin (2 μg), Tetracycline (30 μg)
*E. coli*	Ampicillin (10 μg), Amoxicillin-clavulanic acid (20/10 μg), Ceftiofur (30 μg), Cefpodoxime (10 μg), Sulfamethoxazole-trimethoprim (23.75/1.25 μg), Enrofloxacin (5 μg), Kanamycin (30 μg), Streptomycin (10 μg),Tetracycline (30 μg)

## Data Availability

The original data presented in the study are openly available.

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
