# Peer review of "Longitudinal Patterns in the Isolation and Antimicrobial Resistance of Bovine Mastitis-Causing Bacteria in Ireland"

_antibiotics, 2025, doi:10.3390/antibiotics14030243_

Round 1
Reviewer 1 Report
Comments and Suggestions for Authors
General Comments: The paper titled "Longitudinal Patterns in the Isolation and Antimicrobial Resistance of Bovine Mastitis-Causing Bacteria in Ireland" presents the results of mastitis isolation and antimicrobial sensitivity testing patterns between 2020 and 2023 of Bovine Mastitis-Causing Bacteria from selected farms in Ireland. Overall, the paper is well-written and may provide a valuable contribution to the understanding of the frequency of isolation and antimicrobial resistance of Staphylococcus aureus, Streptococcus uberis, and Escherichia coli. However, there are critical areas that need further clarification to provide the reader with a better understanding of the study and the potential limitations in interpreting the results:
Introduction: The description of the objectives of the study is unclear (Ln 70-76). For example, Ln 70-71 ("This study aimed…and control measures.") does not describe the objectives; instead, it only states the potential implication of the results obtained. Authors need to provide a clear description of the objectives (Suggestion: "This study aimed to evaluate the seasonality of the frequency of isolation, phylogenetic characteristics, and antimicrobial resistance of mastitis-causing pathogens, based on data collected by the Department of Agriculture, Food, and the Marine's Regional Veterinary Laboratories (RVL)").
Methods: The methods section is not sufficiently detailed, especially in terms of the farm description from which samples were collected. Are 7,833 samples representative of how many herds? What were the criteria for sample collection, clinical or subclinical mastitis? Are herds representative of what is claimed in the title (...of Ireland)? Additionally, a better description of the statistical analysis is needed (Ln 416).
Results: In Figure 1, it is not clear the pattern of season according to the mastitis pathogen (Ln 90). Is there any statistical significance among the years and pathogens?
Discussion: Ln 204-216 are not relevant for discussion, as it only brings some literature review about the topic but does not discuss the results. Authors should also emphasize the limitations of the results concerning the representativeness of isolates.
Conclusions: Ln 494 describes a general conclusion, which is not clearly supported by the results of Figure 1.
Minor Review
Ln 13-16: The description of the objectives of the study is unclear. Additionally, the abstract should be reviewed based on the general comments.
Ln 87: Provide a description of the abbreviation: RVLs.
Ln 117-118: This information should be provided in the MM section.
Ln 319: It is too vague (which specific methods?)
Ln 319-320: “Several…mastitis infections” is not relevant for the MM section.
Ln 328-331: It is not clear how (criteria) and for which specific isolates the MALDI-TOF or the API methodology were used. Were all isolates submitted to MALDI-TOF identification?
Ln 337: Which specific criteria were used to select “relevant isolates”?
Ln 350: Serosep: Provide more information.
Ln 370: Which specific criteria were used to select “relevant resistant patterns”?
Author Response
Dear Reviewer,
We sincerely appreciate the time and effort you have dedicated to reviewing my manuscript. Your insightful comments and suggestions have significantly improved the clarity and quality of the paper.
Please find below our detailed responses to all your comments, addressing the points you raised. I hope that the revisions made align with your expectations and further enhance the manuscript.
Thank you once again for your valuable feedback.
Reviewer 1 queries
Introduction:
Question:
The description of the objectives of the study is unclear (Ln 70-76). For example, Ln 70-71 ("This study aimed…and control measures.") does not describe the objectives; instead, it only states the potential implication of the results obtained. Authors need to provide a clear description of the objectives (Suggestion: "This study aimed to evaluate the seasonality of the frequency of isolation, phylogenetic characteristics, and antimicrobial resistance of mastitis-causing pathogens, based on data collected by the Department of Agriculture, Food, and the Marine's Regional Veterinary Laboratories (RVL)").
Answer
Thank you for suggesting an improvement to the introduction. The suggested text has been added to the manuscript and the previous one deleted.
Methods:
Question
The methods section is not sufficiently detailed, especially in terms of the farm description from which samples were collected. Are 7,833 samples representative of how many herds? What were the criteria for sample collection, clinical or subclinical mastitis? Are herds representative of what is claimed in the title (...of Ireland)? Additionally, a better description of the statistical analysis is needed (Ln 416).
Answer
The samples were sent to the RVL for diagnostic purposes from 1,762 different dairy farms, about 10% of the total dairy farms in Ireland. This information was now included in the text. Given these are diagnostic samples, the following paragraph was included in the discussion on the first draft of the manuscript: ‘This study relies on diagnostic samples submitted voluntarily to veterinary laboratories, which may be a limitation regarding the representativeness of the findings. Diagnostic samples are typically submitted when clinical symptoms are evident, and as such, they may over-represent cases with more severe or obvious manifestations of mastitis, potentially biasing the pathogen prevalence data. Additionally, farms with higher mastitis awareness or better access to veterinary services may be more likely to submit samples, further affecting the generalisability of the results. Therefore, the findings should be interpreted with caution.’ This text has now been moved to the first section of material and methods.
Criteria for sample testing information was added: ‘Samples were tested if deemed appropriate by laboratory analysts. The criteria followed for accepting a sample was: (i) sample has been obtained from an individual animal (i.e. is not a bulk tank milk (BTM) sample) (ii) sample does not contain visible dirt or faecal material on receipt to the lab, (iii) sample does not contain preservative tablets for somatic cell count (SCC), and (iv) sample has been collected in an appropriate container (i.e. a sterile tube or pot). All samples were tested using the California mastitis test (CMT) as a preliminary tool to confirm inflammation of the udder. Microbiological culture and antimicrobial susceptibility testing was performed on samples with scores: trace, +1, +2 and +3, which can detect SCC from 150,000 – 500,000 (trace) to >5 million (+3) [40].’
In addition, more information with regards to software employed and further statistical analysis were included: ‘ Given the nature of the data, lower levels of submissions were received in certain months. Using a 90% confidence level and 10% margin of error, with a prevalence of 20% (for the three most common bacterial species), the minimum sample size should be 43 samples using the formula for sample size in proportion studies [46]. Final average monthly sample numbers were 64 in 2020, 58 in 2021, 78 in 2022 and 71 in 2023. Months where the number of samples were below 43 were December and January in 2020, 2021, 2022 and 2023, and September in 2020. For this purpose, seasons were classified as; Winter: January to March, Spring: April to June, Summer: July to September and Autumn: October to December. Although traditional Irish seasons vary from this classification, December and January are months in which lower samples are generally received by the diagnostic laboratories, as we have observed. This can introduce variability and reduce the statistical power of the analysis, potentially biasing seasonal trends; to mitigate this, these months were distributed across different seasons (e.g., Winter: January-March, Autumn: October-December) to ensure their effects are balanced within the seasonal averages. This approach ensures that each season is represented by an adequate sample size, making the analysis more robust and reliable. In addition, power calculation by season using a significance level of 0.05 and an effect size of 0.2 and the two-proportion Z-test method was performed. This effect size reflects a meaningful and biologically relevant difference in bacterial prevalence.
The intensity of seasonal occurrence was estimated using a direct calculation of the peak-to-low ratio and corresponding 95% confidence intervals. Seasonal averages were calculated by grouping monthly incidence percentages into seasons. For each species and year, the average incidence percentage within each season was calculated. A population size per season of 204 was used, because the average total number of isolates per month was 68. A fixed population size per season was used to reflect the proportion of each species in the relation to the overall dataset and to ensure that comparisons across seasons and years are standardized. Peak-to-low ratios were calculated simply by dividing the highest seasonal average by the lowest. Peak-to-low ratio were not performed for S. dysgalactiae data, as average seasonal isolate numbers were very low (n=16) and ratios may not reflect meaningful seasonal trends.
Seasonal average data for all bacterial species and by individual species were tested for normality using the Shapiro-Wilk normality test, and for homogeneity of variance using Bartlett test of homogeneity of variances. ANOVA was used to test for differences between seasons and bacterial species if assumptions were met. Tukey multiple comparisons of means was used as post-hoc test. Otherwise, Kruskal-Wallis rank sum test or generalised linear models (GLM) with a quasibinomial family, followed by Tukey's Honest Significant Difference (HSD) test for post-hoc comparisons for more complex comparisons were employed to test for differences if the data did not meet normality and homogeneity assumptions. P values of <0.05 were considered to be statistically significant.
Data management and statistical analyses were performed using R version 4.3 (packages: FSA_0.9.6, tidyr_1.3.1, ggplot2_3.5.1, emmeans_1.10.6 dplyr_1.1.4 and readxl_1.4.3) and Microsoft Excel version 16.85. ‘
Results:
Question
In Figure 1, it is not clear the pattern of season according to the mastitis pathogen (Ln 90). Is there any statistical significance among the years and pathogens?
Answer:
Figure 1 has been edited to show each seasonal proportions for bacterial species individually for more clarity.
Statistical analyis was performed, and description included in Materials and methods section, as indicated in previous answer. The results of this analysis have been included in a new table 2, and a summary in the text: ‘ GLM with Tukeys HSD were employed to analyse overall seasonal differences between species (Table 2). S. uberis and S. aureus were the only pair showing no significant differences in any season. E. coli did not show significant differences in Spring with S. aureus or S. uberis. All other comparisons were significantly different.’
Discussion:
Question
Ln 204-216 are not relevant for discussion, as it only brings some literature review about the topic but does not discuss the results. Authors should also emphasize the limitations of the results concerning the representativeness of isolates.
Answer: The first paragraph from the discussion is a brief introduction, it has now been edited to discuss the results:’… However, this poses new challenges, particularly in herds with suboptimal mastitis control, and a better understanding of the epidemiology of mastitis pathogens present in a particular farm is therefore key [16]. We found that the seasonality patterns and laboratory results obtained in previous years on the same herd, along with clinical signs, can be useful tools for PVPs to decide on appropriate therapeutic treatment of mastitis cases. This is particularly important in acute or severe cases where action is needed under animal welfare grounds before any laboratory results can be obtained.’
Limitations were included in the last paragraph of the discussion. They have now been moved to the sample source section of materials and methods: ‘This study relies on diagnostic samples submitted voluntarily to veterinary laboratories, which may be a limitation regarding the representativeness of the findings. Diagnostic samples are typically submitted when clinical symptoms are evident, and as such, they may over-represent cases with more severe or obvious manifestations of mastitis, potentially biasing the pathogen prevalence data. Additionally, farms with higher mastitis awareness or better access to veterinary services may be more likely to submit samples, further affecting the generalisability of the results. Therefore, the findings should be interpreted with caution.’
Conclusions:
Question
Ln 494 describes a general conclusion, which is not clearly supported by the results of Figure 1.
Answer: Figure one has now been edited to show more clearly these results, and statistical analysis was made to support this.
Minor Review
Ln 13-16: The description of the objectives of the study is unclear. Additionally, the abstract should be reviewed based on the general comments.
Answer:Done
Ln 87: Provide a description of the abbreviation: RVLs.
Answer: Done
Ln 117-118: This information should be provided in the MM section.
Answer: Done
Ln 319: It is too vague (which specific methods?).
Answer: They are further explained in the paragraph.
Ln 319-320: “Several…mastitis infections” is not relevant for the MM section.
Answer: This was removed
Ln 328-331: It is not clear how (criteria) and for which specific isolates the MALDI-TOF or the API methodology were used. Were all isolates submitted to MALDI-TOF identification?
Answer: Isolates were inintially tested by API. If any resistance at all was observed, they were further identified by MALDI and tested by broth microdilution. This was added to the relevant methods section.
Ln 337: Which specific criteria were used to select “relevant isolates”?
Answer: This was changed to ‘on all pure isolates obtained’.
Ln 350: Serosep: Provide more information. Done
Ln 370: Which specific criteria were used to select “relevant resistant patterns”?
Answer: In fact, isolates showing any resistance were tested. This has been changed in the text.
Reviewer 2 Report
Comments and Suggestions for Authors
Author Response
Dear Reviewer,
We sincerely appreciate the time and effort you have dedicated to reviewing my manuscript. Your insightful comments and suggestions have significantly improved the clarity and quality of the paper.
Please find below our detailed responses to all your comments, addressing the points you raised. I hope that the revisions made align with your expectations and further enhance the manuscript.
Thank you once again for your valuable feedback.
Reviewer 2 queries
- The article provies important data on the seasonal patterns and antimicrobial resistance patterns of mastitis-causing bacteria in Ireland. The authors are advised to further emphasize the importance of these findings for developing targeted biosafety measures and monitoring programs, as well as how they can help reduce the use of antibiotics in the Irish dairy industry.
Answer: Additional text was added to the discussion to covert this query: ‘…we believe these results are relevant from a biosecurity point of view; herds with consistent mastitis problems during the summer months are advised to control faecal contamination of their pastures, thereby reducing the presence of S. uberis. This can be achieved by rotating grazing areas, avoiding overstocking, and ensuring cows have access to dry resting areas. Routine maintenance of the milking machine, including checking vacuum levels and replacing worn components, and implementing effective post-milking disinfection protocols, such as the use of teat dips with proven efficacy, will also help reduce the incidence of mastitis caused by S. aureus [15,31]. On the other hand, herds experiencing mastitis issues during the Winter are more likely affected by E. coli and are advised to focus on maintaining clean and dry bedding, ensuring proper ventilation in housing facilities, and minimizing overcrowding [18]. Peri-parturient stress, which often occurs around calving due to metabolic and physiological changes, can compromise the immune system, increasing susceptibility to infections. To mitigate this, herds should prioritize stress reduction strategies, such as providing sufficient space, minimizing disruptions, and ensuring high-quality nutrition during the transition period [32].’
The following paragraph was also included: ‘It is crucial to perform culture and antimicrobial sensitivity testing of milk samples to guide targeted therapy and reduce the unnecessary use of antibiotics. However, in acute cases of mastitis, where immediate intervention is required, treatment often needs to be initiated empirically before laboratory results become available. In such situations, the choice of antimicrobial should be guided by historical farm-level sensitivity patterns and knowledge of likely pathogens, while laboratory confirmation remains essential to refine ongoing treatment and inform future decisions. Having into account the seasonality and AMR findings from this study can be useful tools for PVP dealing with severe mastitis cases. For instance, a mastitis infection during the summer months is more likely caused by S. uberis or S. aureus. Therapy can therefore be better informed with a more targeted approach. In addition, if this infection is not responsive to therapeutic treatment with penicillin or other early generation cephalosporin, the etiological agent will most likely be S. aureus.’
- The authos are advised to provide more details on sample size calculations and discuss whether there is sufficient statistical power to detect differences in antimicrobial resistance over the years.
The following was added to the text: ‘In addition, power calculation by season using a significance level of 0.05 and an effect size of 0.2 and the two-proportion Z-test method was performed. This effect size reflects a meaningful and biologically relevant difference in bacterial prevalence.’. Calculations indicated that statistical power for the seasonal calculations were >0.8. With regards to the antimicrobial resistance sections, this was a descriptive analysis of those farms that had repeatedly submitted samples for E. coli, S. aureus, and S. uberis over multiple years. The sample sizes (55 for E. coli, 76 for S. aureus, and 92 for S. uberis) represent the available data for herds with recurring issues between 2020 and 2023. Therefore, no power calculations were performed. This information was added to the text: ‘ A descriptive analysis of the resistance patterns found in these herds was performed.’
- The study observed seasonal patterns of different pathogens. The authors are advised to further explore the possible reasons for these patterns, including climate, management practices, and bacterial characteristics.
While we included comparison to additional studies that can explain these patterns, we have added a paragraph where we acknowledge that our study does not include data on management or herd characteristics, which are important factors affecting mastitis development.
- The article mentions resistance genes and virulence factors but does not discuss their association in detail. The authors are advised to further analyze how these factors might jointly affect the pathogenesis and treatment of mastitis
The aim of this section of the study was to analyze whether any virulence factors or antimicrobial resistance (AMR) genes were associated with the persistence of a strain within a herd. This association was tested, and no significant relationships were identified between the presence of virulence genes or AMR genes and strain persistence. Furthermore, it is important to note that we did not collect data on the specific pathogenesis of each individual mastitis case. Therefore, while our results provide valuable insights into the genomic characteristics of the strains, they do not allow us to explore how these factors may interact to influence pathogenesis or treatment outcomes. added The following was added to the discussion to complete information about virulence genes: We found the presence of several genes related to immune evasion (cap8, isd, lukD), tissue damage and inflammation (hld, hla, hlg), iron acquisition (isd system) or bacterial persistence (aur, sspB, sspC) that are key for the pathogenesis of S. aureus infections [36]. However, virulence genes that were associated with persistent infections according to Haveri et al., (2007) (sed and sej) were not present in the strains from this study….
- The article provides data from 2020 to 2023. The authors are advised to discuss how these findings can be incorporated into long-term monitoring plans and consider how to regularly update these data to track trends in antimicrobial resistance.
The following text has been added to the discussion: ‘This study utilizes data from routine diagnostic submissions which are used as a source of passive surveillance for antimicrobial resistance (AMR) in mastitis pathogens in Ireland. Antimicrobial sensitivity results are published annually in the All Island Disease Surveillance Report, which can be found in https://www.animalhealthsurveillance.agriculture.gov.ie. While there is currently no legislation mandating the monitoring of AMR in clinical veterinary isolates, several initiatives, such as JAMRAI 2 and ENOVAT, are working towards standardizing and harmonizing testing and reporting protocols across the EU. These efforts aim to develop a comprehensive EU-wide AMR surveillance program. Importantly, the data generated from this testing will contribute to JAMRAI 2, providing valuable insights into resistance trends and supporting the development of such frameworks.’
- Figure 1 illustrates the seasonality of bacterial isolates from mastitis milk samples between 2020 and 2023. Please provide further explanation on how the months with sample numbers below 43 were determined and discuss the potential impact this might have on the analysis of seasonal patterns.
The following was added to the text: ‘Months where the number of samples were below 43 were December and January in 2020, 2021, 2022 and 2023, and September in 2020. For this purpose, seasons were classified as; Winter: January to March, Spring: April to June, Summer: July to September and Autumn: October to December. Although traditional Irish seasons vary from this classification, December and January are months in which lower samples are generally received by the diagnostic laboratories, as we have observed. This can introduce variability and reduce the statistical power of the analysis, potentially biasing seasonal trends; to mitigate this, these months were distributed across different seasons (e.g., Winter: January-March, Autumn: October-December) to ensure their effects are balanced within the seasonal averages. This approach ensures that each season is represented by an adequate sample size, making the analysis more robust and reliable. A significance level of 0.05 and an effect size of 0.2 were chosen for power calculations using the two-proportion Z-test method, as they reflect a meaningful and biologically relevant difference in bacterial prevalence, which is approximately 20% for the species of interest.’
- The species and genus names in Table 1 and 2 should be in italics.
Answer: Done
Reviewer 3 Report
Comments and Suggestions for Authors
MDPI Antibiotics: “Longitudinal patterns in the isolation and antimicrobial resistance of bovine mastitis-causing bacteria in Ireland” by Amalia Naranjo-Lucena et al.
This manuscript investigates the seasonality and antimicrobial resistance of bovine mastitis isolates from Ireland over the period of 2020-2023. Isolates considered included Escherichia coli, Staphylococcus aureus, Streptococcus uberis and Streptococcus dysgalactiae. Isolates were obtained from milk samples from clinical and subclinical bovine mastitis, based upon submissions to the Department of Agriculture Food and the Marines diagnostic laboratories. Milk samples were cultured and data on the pathogens studied was collected and analyzed, along with Whole Genome Sequencing (WGS) of St. aureus isolates. The results showed clear seasonality in terms of occurrence of mastitis caused by E. coli, St. aureus and Str. uberis. Looking at the persistence of resistance by isolates to several antibiotics, the authors observed highest persistence of St. aureus against penicillin and Str. uberis against pirlimycin. Sequencing of St. aureus isolates revealed that strains causing mastitis were similar genetically in different years for isolates in nearly one-half of herds. The authors concluded that “These findings underscore the importance of implementing targeted biosecurity measures and monitoring programs to mitigate the spread of mastitis-causing pathogens and enhance antimicrobial stewardship…”
The study was designed appropriately and used methods effective to answer the questions the authors asked. The manuscript is presented in a readable and understandable manner, with few errors or misspellings. Mastitis is a common and costly disease and further insight into the epidemiology of mastitis can contribute to improved treatment and control.
The authors could improve the manuscript by making a stronger and more specific case in terms of how the findings of their study “support the development of improved prevention and control measures.”
Specific comments for the authors to consider include:
Introduction, pp. 1-2, ll. 39-42: This sentence would read better if the last part of the sentence was moved to the front of the sentence. Thus, start with: “The dairy industry remains a cornerstone of Irish agriculture despite….”
Introduction, p. 2, l. 57: Please consider replacement of the word “consumption” in this sentence with “use” or “utilization.”
Introduction, p. 2, ll. 64-66: Presumably, the reference for the sentence beginning with “The need to reduce…” and including reference to “RONAFA Joint Scientific….” Is #11. I would cite it after this sentence.
Results, p. 3, Table 1: Several issues are present with this table. I believe there are errors in the %’s given under the column for 2022—please check. It is not very intuitive where the percentages in the columns come from. Could you, in some way, give the reader a little more insight into that. Perhaps you could include a notation that the percentages are of the total number of isolations per year under “Seasonality.” Or you could have a footnote to the table: “Percentages of each bacteria per year are calculated based on a total of 767 for 2020, 701 for 2021, etc.” Or, you may have other ideas for how to do it.
Results, p. 5, ll. 117-120: I am still trying to get an understanding of the numbers here. Are you saying that of the total of 190 herds, only 40 of them had more than one pathogen detected as a cause of mastitis over the 4 years? There were 7,833 samples collected from 190 herds over 4 years. That would be an average of 7,833 samples/4 years=1958 samples/year. Then 1958 samples/190 herds=10 samples per average herd per year. Perhaps, just a little more detail on the herds and results would help. Like, could you say what the approximate percentage of samples you got a bacteriologic result/diagnosis and maybe average size and range of herd sizes?
Results, p. 5, ll. 122-136: You should be consistent in terms of reporting values in terms of significant figures after the decimals. For some, you report a whole number: 49%, 32% on l. 132; for others, you report to 1 or 2 significant figures: 30.9% and 25.45% on l. 126.
Results, p. 5, ll. 145-150 and p. 6, Table 4, l. 151: Table 4 likely needs some revision, and the paragraph and Table can be modified to be more clear. The title includes an error, “from” is misspelled as “rom.” On ll. 129-130, you refer to levels SLC1 to SLC24 being indicated in Table. If you are referring to the headings SLC1, SLC5, SLC7, and SLC24, then it isn’t SLC1 to SLC24. The table needs to indicate what SLC and AD are (or explain in footnote). What are the numbers under the columns headed by SLC1, 5, 7 and 24?
Results, Figure 2, p. 7, ll. 168-171: This figure is quite faint and hard to read. The colors and the bold numbers show well. A little more explanation may also help.
M&M, p. 11, l. 322 (and after): It is customary that the locations of the companies supplying M&M is usually given following the names. Example: SAS, Inc., Cary, NC)
M&M, p. 11, l. 335: Word missing (isolated?) in this portion of sentence: “….most commonly bacteria were…”
M&M, p. 12, Table 8: Under drug list for E. coli, no disc size is given for Streptomycin.
M&M, p. 12, l. 348: sensititre needs capital S: Sensititre.
M&M, p. 12, l. 372: Byrne et al., 2024 is Reference 41.
M&M, p. 13, l. 397: Ratio needs to be plural: ratios.
M&M, p. 14, l. 438: diagnostics needs to be capitalized: Diagnostics.
M&M, p. 15, l. 469: Word missing here: “..sequencing reads greater than…” Should be “..sequencing reads were greater than…”
M&M, p. 15, ll. 471-472: I think that this should be “The fastq files for each sample were assembled..” Next line: Not sure what the “()” is after “Spades Version.”
M&M, p. 15, ll. 475-476: The following references are not cited by numbers: Leopold et al., 2014; Chen et al., 2016; Feldgarden et al., 2019; Mellmann et al., 2007.
M&M, p. 15, l. 477: “…profiles of the samples were determined…”
Abbreviations, p. 16, l. 523 and following:
Is there a rationale for the order in which the abbreviations are listed? Order of appearance? Other? AD should be included as an abbreviation.
Please check the following references for completeness, missing information, etc.:
#2: Book or journal source not given
#6: Same as above.
#8: Same as above.
#9: Same as above.
#10: What is JDSC?
#11: Incomplete reference.
#12: Edition misspelled.
#13: Incomplete reference information.
#31: Incomplete reference.
#32: Incomplete reference.
Author Response
Dear reviewers 3,
We sincerely appreciate the time and effort you have dedicated to reviewing my manuscript. Your insightful comments and suggestions have significantly improved the clarity and quality of the paper.
Please find below our detailed responses to all your comments, addressing the points you raised. I hope that the revisions made align with your expectations and further enhance the manuscript.
Thank you once again for your valuable feedback.
Reviewer 3 queries
The study was designed appropriately and used methods effective to answer the questions the authors asked. The manuscript is presented in a readable and understandable manner, with few errors or misspellings. Mastitis is a common and costly disease and further insight into the epidemiology of mastitis can contribute to improved treatment and control.
The authors could improve the manuscript by making a stronger and more specific case in terms of how the findings of their study “support the development of improved prevention and control measures.”
Answer: Additional text was added to the discussion to covert this query: ‘…we believe these results are relevant from a biosecurity point of view; herds with consistent mastitis problems during the summer months are advised to control faecal contamination of their pastures, thereby reducing the presence of S. uberis. This can be achieved by rotating grazing areas, avoiding overstocking, and ensuring cows have access to dry resting areas. Routine maintenance of the milking machine, including checking vacuum levels and replacing worn components, and implementing effective post-milking disinfection protocols, such as the use of teat dips with proven efficacy, will also help reduce the incidence of mastitis caused by S. aureus [15,31]. On the other hand, herds experiencing mastitis issues during the Winter are more likely affected by E. coli and are advised to focus on maintaining clean and dry bedding, ensuring proper ventilation in housing facilities, and minimizing overcrowding [18]. Peri-parturient stress, which often occurs around calving due to metabolic and physiological changes, can compromise the immune system, increasing susceptibility to infections. To mitigate this, herds should prioritize stress reduction strategies, such as providing sufficient space, minimizing disruptions, and ensuring high-quality nutrition during the transition period [32].’
The following paragraph was also included: ‘It is crucial to perform culture and antimicrobial sensitivity testing of milk samples to guide targeted therapy and reduce the unnecessary use of antibiotics. However, in acute cases of mastitis, where immediate intervention is required, treatment often needs to be initiated empirically before laboratory results become available. In such situations, the choice of antimicrobial should be guided by historical farm-level sensitivity patterns and knowledge of likely pathogens, while laboratory confirmation remains essential to refine ongoing treatment and inform future decisions. Having into account the seasonality and AMR findings from this study can be useful tools for PVP dealing with severe mastitis cases. For instance, a mastitis infection during the summer months is more likely caused by S. uberis or S. aureus. Therapy can therefore be better informed with a more targeted approach. In addition, if this infection is not responsive to therapeutic treatment with penicillin or other early generation cephalosporin, the etiological agent will most likely be S. aureus.’
Specific comments for the authors to consider include:
Introduction, pp. 1-2, ll. 39-42: This sentence would read better if the last part of the sentence was moved to the front of the sentence. Thus, start with: “The dairy industry remains a cornerstone of Irish agriculture despite….”
Answer: Done
Introduction, p. 2, l. 57: Please consider replacement of the word “consumption” in this sentence with “use” or “utilization.”
Answer: Done
Introduction, p. 2, ll. 64-66: Presumably, the reference for the sentence beginning with “The need to reduce…” and including reference to “RONAFA Joint Scientific….” Is #11. I would cite it after this sentence.
Answer: Done
Results, p. 3, Table 1: Several issues are present with this table. I believe there are errors in the %’s given under the column for 2022—please check. It is not very intuitive where the percentages in the columns come from. Could you, in some way, give the reader a little more insight into that. Perhaps you could include a notation that the percentages are of the total number of isolations per year under “Seasonality.” Or you could have a footnote to the table: “Percentages of each bacteria per year are calculated based on a total of 767 for 2020, 701 for 2021, etc.” Or, you may have other ideas for how to do it.
Answer: This table represents the annual proportion of each bacterial species. The description has been edited to state: ‘(results show percentage within total isolates obtained per year, including minor species, and number of isolates in parenthesis)’ and the total annual isolates were also added for each year.
Results, p. 5, ll. 117-120: I am still trying to get an understanding of the numbers here. Are you saying that of the total of 190 herds, only 40 of them had more than one pathogen detected as a cause of mastitis over the 4 years? There were 7,833 samples collected from 190 herds over 4 years. That would be an average of 7,833 samples/4 years=1958 samples/year. Then 1958 samples/190 herds=10 samples per average herd per year. Perhaps, just a little more detail on the herds and results would help. Like, could you say what the approximate percentage of samples you got a bacteriologic result/diagnosis and maybe average size and range of herd sizes?
Answer: Unfortunately, we do not have metadata on characteristics or management of the dairy herds. The following text was included to further clarify that these 190 were a cohort from the total number of 1,762 herds that sent samples. These 190 herds submitted a sample in two or three different years, which were appropriate to be tested and resulted in the pure culture of one of the four main isolates discussed in this study. It is for this reason that the number of herds is reduced so much.
‘For this analysis, only those herds that had samples submitted on two or more years that were appropriate to be tested, and from which a pure culture of E. coli, S. aureus, S. uberis or S. dysgalactiae was obtained were selected. This resulted in a cohort of 190 herds out of the 1,655 that submitted samples during that time. In 40 of these herds, more than one pathogen was detected as the cause of mastitis. E. coli was isolated from 55 herds, S. aureus from 76, S. uberis from 92 and S. dysgalactiae from seven herds. Given the descriptive nature of this analysis, no statistical power calculation or inferential analysis was conducted.
.’
Results, p. 5, ll. 122-136: You should be consistent in terms of reporting values in terms of significant figures after the decimals. For some, you report a whole number: 49%, 32% on l. 132; for others, you report to 1 or 2 significant figures: 30.9% and 25.45% on l. 126.
Answer: Done
Results, p. 5, ll. 145-150 and p. 6, Table 4, l. 151: Table 4 likely needs some revision, and the paragraph and Table can be modified to be more clear. The title includes an error, “from” is misspelled as “rom.” On ll. 129-130, you refer to levels SLC1 to SLC24 being indicated in Table. If you are referring to the headings SLC1, SLC5, SLC7, and SLC24, then it isn’t SLC1 to SLC24. The table needs to indicate what SLC and AD are (or explain in footnote). What are the numbers under the columns headed by SLC1, 5, 7 and 24?
Answer: The typo was corrected in the table title. The paragraph now better explains the information exposed. We acknowledge the discrepancy in the numbers presented in the table. After re-evaluating the data, we identified and corrected the errors. The revised table now accurately reflects the correct values of allelic differences between isolates within each MST cluster. In addition, the text in M&M has been edited to: ‘The software generated single linkage clusters using thresholds of 1, 5, 7 and 24 allelic differences (AD).’; in the results text: ‘Four different thresholds of single linkage cluster (SLC) were determined and number of allelic differences according to the levels SLC1, SLC5, SLC7 and SLC24 are indicated in table 5.’; and in the table title: ‘Core genome MLST, spa typing results of 43 S. aureus isolates from 21 herds and MST clusters with maximum allelic differences found within each SLC threshold.’ Additionally, an MST cluster column was added to the table containing cluster ID.
Results, Figure 2, p. 7, ll. 168-171: This figure is quite faint and hard to read. The colors and the bold numbers show well. A little more explanation may also help.
Answer: The sample numbers have been removed from the figure for visualization purposes and the image quality has been improved.
M&M, p. 11, l. 322 (and after): It is customary that the locations of the companies supplying M&M is usually given following the names. Example: SAS, Inc., Cary, NC)
Answer: Done
M&M, p. 11, l. 335: Word missing (isolated?) in this portion of sentence: “….most commonly bacteria were…”
Answer: Done
M&M, p. 12, Table 8: Under drug list for E. coli, no disc size is given for Streptomycin.
Answer: Done
M&M, p. 12, l. 348: sensititre needs capital S: Sensititre.
Answer: Done
M&M, p. 12, l. 372: Byrne et al., 2024 is Reference 41.
Answer: Done
M&M, p. 13, l. 397: Ratio needs to be plural: ratios.
Answer: Done
M&M, p. 14, l. 438: diagnostics needs to be capitalized: Diagnostics.
Answer: Done
M&M, p. 15, l. 469: Word missing here: “..sequencing reads greater than…” Should be “..sequencing reads were greater than…”
Answer: Done
M&M, p. 15, ll. 471-472: I think that this should be “The fastq files for each sample were assembled..” Next line: Not sure what the “()” is after “Spades Version.”
Answer: corrected
M&M, p. 15, ll. 475-476: The following references are not cited by numbers: Leopold et al., 2014; Chen et al., 2016; Feldgarden et al., 2019; Mellmann et al., 2007.
Answer: corrected
M&M, p. 15, l. 477: “…profiles of the samples were determined…”
Answer: Done
Abbreviations, p. 16, l. 523 and following:
Is there a rationale for the order in which the abbreviations are listed? Order of appearance? Other? AD should be included as an abbreviation.
Answer: Completed and arranged by alphabetical order
Please check the following references for completeness, missing information, etc.:
Answer: The references instructions to authors does not include how to reference reports, so I have added a comment at the end of each one of them.
#2: Book or journal source not given
Answer: This is a report by a national agency. Their name has been added to the reference.
#6: Same as above.
Answer: This is an international report, by 3 institutions. Added the ‘International Report’ at the end.
#8: Same as above.
Answer: This is an international report, by 3 institutions. Added the ‘International Report’ at the end.
#9: Same as above.
Answer: This is an international report, by 3 institutions. Added the ‘International Report’ at the end.
#10: What is JDSC?
Answer: Journal of Dairy Science Communications, added to reference.
#11: Incomplete reference.
Answer: Completed
#12: Edition misspelled.
Answer: Corrected
#13: Incomplete reference information.
Answer: This is a report by a national agency
#31: Incomplete reference.
Answer: This is a report by a national agency, added ‘Report’
#32: Incomplete reference.
Answer: This is a report by a national agency, added ‘Report’
Round 2
Reviewer 2 Report
Comments and Suggestions for Authors
Authors have revised the manuscript and it can be accepted.
Author Response
Dear reviewer,
Thank you for considering our improvements of the manuscript for further publication.
Kind regards,
Amalia Naranjo Lucena